# Bioinformatics Analysis and Validation of Potential Markers Associated with Prediction and Prognosis of Gastric Cancer

**DOI:** 10.3390/ijms25115880

**Published:** 2024-05-28

**Authors:** Tasuku Matsuoka, Masakazu Yashiro

**Affiliations:** 1Department of Molecular Oncology and Therapeutics, Osaka Metropolitan University Graduate School of Medicine, 1-4-3 Asahi-machi, Abeno-ku, Osaka 5458585, Japan; t22738q@omu.ac.jp; 2Institute of Medical Genetics, Osaka Metropolitan University, 1-4-3 Asahi-machi, Abeno-ku, Osaka 5458585, Japan

**Keywords:** gastric cancer, prognosis, prediction, biomarker, bioinformatics, multi-omics, machine learning

## Abstract

Gastric cancer (GC) is one of the most common cancers worldwide. Most patients are diagnosed at the progressive stage of the disease, and current anticancer drug advancements are still lacking. Therefore, it is crucial to find relevant biomarkers with the accurate prediction of prognoses and good predictive accuracy to select appropriate patients with GC. Recent advances in molecular profiling technologies, including genomics, epigenomics, transcriptomics, proteomics, and metabolomics, have enabled the approach of GC biology at multiple levels of omics interaction networks. Systemic biological analyses, such as computational inference of “big data” and advanced bioinformatic approaches, are emerging to identify the key molecular biomarkers of GC, which would benefit targeted therapies. This review summarizes the current status of how bioinformatics analysis contributes to biomarker discovery for prognosis and prediction of therapeutic efficacy in GC based on a search of the medical literature. We highlight emerging individual multi-omics datasets, such as genomics, epigenomics, transcriptomics, proteomics, and metabolomics, for validating putative markers. Finally, we discuss the current challenges and future perspectives to integrate multi-omics analysis for improving biomarker implementation. The practical integration of bioinformatics analysis and multi-omics datasets under complementary computational analysis is having a great impact on the search for predictive and prognostic biomarkers and may lead to an important revolution in treatment.

## 1. Introduction

Gastric cancer (GC) is a common cancer type, with more than one million new cases each year [1]. Although recent developments in endoscopy, imaging, surgical procedures, and the use of anti-cancer drugs have somewhat prolonged the overall outcome of GC patients, the 5-year survival of patients with advanced GC remains <30%, mainly due to the lack of efficient therapies [2]. So far, comprehensive molecular analysis has identified many molecular alterations and dysregulated pathways in GC. Mutations in the *TP53* and *CDH1*/RHOA genes are well-known [3]. Concerning copy number alterations, the amplification of genes involved in tyrosine kinase receptor pathways, such as *FGFR2*, *HER2*, *EGFR*, *MET*, and *WNT*, has been noted, as well. Other important mutations in GC involve the *ARID1A*, *PIK3CA*, CCNE1, CLDN18-ARHGAP26, and *BRCA2* genes [4]. Despite these advances in knowledge only one oncogenic driver-based therapy (Her2-directed treatment with trastuzumab) has shown acceptable survival benefit [5]. Other therapeutic options, such as cytotoxic agents or anti-angiogenic therapy, are only transiently useful except checkpoint inhibitors for patients with microsatellite unstable GCs, which are infrequent at the advanced stage. Therefore, it is crucial to find relevant biomarkers with good predictive accuracy to select those patients with GC who are likely to benefit from targeted therapies.

The development of innovative computational approaches for integrating data from various cohorts would be useful for resolving these problems. Rapid technological revolutions in genome-wide sequencing, such as next-generation sequencing (NGS), have recently provided researchers with enormous expression datasets [6]. The advancement of omics data due to breakthroughs in high-throughput technologies has generated the concept of “big data” in cancer, the analysis of which necessitates large computational resources and bioinformatic algorithms [7]. With the interpretation of bioinformatics technology, a comprehensive understanding of genetic, epigenetic, proteomic, transcriptomic, and metabolic processes has been developed as a framework with which researchers can decipher the various mechanisms responsible for the heterogeneity of GC [8]. “Omics” is a series of approaches that aim to characterize and quantify the pools of biomolecules within organisms related to structures, functions, and dynamics. However, individual profiling using a single-omics approach only offers a limited landscape of the complex molecular regulatory axes in malignancies. Thus, integration analysis of multi-omics (multi-layer) data is a crucial step toward a greater understanding of the signatures of different mechanisms of carcinogenesis and, indeed, the mechanisms themselves. The integration of bioinformatic analysis with multi-omics platforms is emerging as a promising tool for elucidating accurate prognosis, prediction of effective therapies, and biomarkers to increase the response rate to targeted treatments in patients with GC [9] (Figure 1).

In this review, we summarize an overview of how bioinformatic analysis contributes to biomarker discovery for prognosis and prediction of therapeutic efficacy in GC. Emerging single-omics datasets, such as genomics, epigenomics, transcriptomics, proteomics, and metabolomics for validating putative markers, are systematically reviewed. Finally, we discuss the current challenges and future perspectives to integrate multi-omics analysis to improve biomarker implementation at the clinical level.

## 2. Genomics-Based Validation of Predictive and Prognostic Markers of GC

Recently, high-throughput technology combined with bioinformatics algorithms has been broadly used in the detection of genetic alteration derived in oncogenesis. To diagnose differentially expressed genes (DEGs) associated with GC, databases such as The Cancer Genome Atlas (TCGA) and Gene Expression Omnibus (GEO) are frequently used [10]. Meanwhile, the former conventional studies have only focused on the distinctive DEGs and disregarded the network complex with a high degree of interaction between the DEGs. Protein–protein interaction (PPI) networks and weighted gene co-expression network analysis (WGCNA) based on the RNA sequencing and microarray data have been revealed to establish potent organized biological strategies for mining the functional gene modules and detecting novel prognostic biomarkers [11,12]. These tools enable a better understanding of the mechanisms underlying the progression of GC and are utilized to identify novel targets for the prediction of effective therapy and prognosis.

While numerous single-gene biomarkers have been described, multi-gene signatures reveal more usefulness and significance for the prognosis of patients with cancer. Current topics of GC predictive and prognostic biomarkers based on a variety of genetic features based on bioinformatics analysis were summarized in Table 1. A recent study integrating data in multiple public databases detected a six-metastasis-related gene (*PCOLCE2*, TMEM132C, *UPK1B*, *SLITRK2*, PM20D1, and *FLJ35024*) set that predicts prognosis in patients with GC through TGCA database analysis. They also evaluated the expression of the six distant-metastasis gens in the GC tissue sample, and the reliability of the prognostic value was shown [13]. Based on 375 TCGA GC samples, 50 DEGs between GC and paired normal tissues were identified [14]. The association of gene expression with immune cells was examined based on the TIMER database. Ten of the fifty genes *(RXRG*, *AGT*, *BCHE*, *UBE2QL1*, *PLCXD3*, *ADCYAP1R1*, *NRCAM*, *MAMDC2*, *CDH19*, and *GAMT*) were linked with the outcome and macrophage infiltration in GC. AGT was correlated to GC carcinogenesis and cooperated with *BRD9*, *GOLPH3*, *NOM1*, *KLHL25*, and *PSMD11* [14]. Interestingly, systematic analysis summarized 39 prognostic gene signatures for GC patients to detect the most overlapped gene. Three genes (*COL1A1*, COL1A2, EGFR) were identified three times, suggesting that they have superior potential in GC prognosis and may provide a theoretic foundation for improving treatment strategies [15]. Another study has identified DEGs by using WGCNA and detected five hub genes (*CEMIP*, CST1, COL8A1, PMEPA1, and *MSLN*) in GC and has constructed a prognostic signature with a good predictive performance by integrating multiple GEO datasets and WGCNA algorithms. Among them, *CEMIP* repressed the growth and migration and impaired the chemoresistance of GC cells to 5-FU, suggesting that *CEMIP* may be applied to precision medicine for GC patients [16]. Similarly, an integrated bioinformatics analysis was performed to identify the common hub genes derived from the PPI and WGCNA network based on the common DEGs from the TCGA-STAD and three GEO datasets (GSE65801, GSE54129, and GSE118916). Among the six common hub genes, *TRYOBP* and *C1QB* were negatively associated with the overall survival (OS) of the GC patients. Both genes were also correlated with the CD8^+^ T cells, CD4^+^ T cells, and macrophages using an immune analysis based on the TCG-STAD [17]. A recent study using a prognostic risk model by multivariate Cox regression analysis in GEO datasets GSE15459 investigated potential prognostic genes [18]. A total of eight genes significantly related to survival were identified, and decreased survival was significantly associated with *AGT*, *SERPINH1*, and *MMP7*, indicating that the abnormal regulation of these genes in cancer may impact the OS of GC. Moreover, the exosome level of AGT and MMP7 were associated with the serum level, and these proteins stimulate the migration of GC cell lines, suggesting that exosomal AGT and MMP7 may be useful biomarkers for GC prognosis and included in GC progression. Notably, a study using DAVID analysis detected clusters of nine genes (*CTHRC1*, *BGN*, *FAP*, *THBS2*, *COL12A1*, *COL5A2*, *SULF1*, *SPP1*, and *COL10A1*) which were associated with alterations in the extracellular matrix (ECM), cell adhesion, and collagen catabolic process. The high expression of BGN, COL5A1, COL10A1, COL12A1, CTHRC1, SULF, and THBS2 are negatively correlated with OS and may serve as useful prognostic biomarkers for GC. Among them, a superior hazard ratio (HR) for poor OS was shown in BGN, THBS2, and CTHRC1 compared with other hub genes [19].

Overall, genetic correlations with GC based on integrated bioinformatic analysis have been identified. The main purpose is to progress our knowledge about GC-associated biological pathways, predicting GC prognosis and treatment outcomes. Moreover, further understanding of the genetic loci associated with GC can allow us to identify more precise predictive/prognostic biomarkers.

## 3. Epigenomics-Based Validation of Predictive and Prognostic Markers of GC

Epigenetic regulations are non-heritable changes induced by covalent tags or other modifications of the genome [20]. Similar to genomics, an increasing number of evidence suggests that epigenetic modifications, including DNA methylation of CpG islands, post-translational modifications of histones, and nucleosome positioning, are also comprised in the GC initiation, progression, and chemo-resistance [21]. DNA methylation alterations are essential to all phases of cancer genomics, and hypomethylation has been revealed to be pivotal in cancer progression [22]. Recently, DNA methylation has been revealed to play a significant predictive role in the prognosis of patients with GC.

The nomogram has been broadly utilized as a predictive method in oncology for a decade [23]. It requires an integrated model and is useful for clinicians to exploit in prognosis prediction. A previous study using a risk score and OS nomogram for predicting the prognosis of GC patients identified six DNA methylation-driven DEGs (*PODN*, *MYO1A*, *NPY*, *MICU3*, *TUBB6*, and *RHOJ*) as prognostic genes using a least absolute shrinkage and selection operator (LASSO) logistic regression model [24]. They verified that hypermethylated five genes (*PODN*, *NPY*, *MICU3*, *TUBB6*, and *RHOJ*) may have a protective role. In contrast, MYO1A was hypomethylated and may be a typical tumor suppressor gene. These six genes may contribute to a cost-effective and accurate prediction of the outcome of GC in clinical use. Another recent study integrated publicly available GC DNA methylation datasets obtained from TCGA and GEO databases to construct a prognostic model by matching differential methylation sites (DMSs) with OS [25]. The prognostic model identified an 11-DMS (CEP290, CCDC69, UBXN8, KDM4A, AKR1B1, RASSF2, KDELR3, CHRNB2, EGR1, ARMC9, and RPN1) signature. Among these DMGs, five DNA methylation driver genes (KDM4A, AKR1B1, RASSF2, CHRNB2, and EGR1) are found to be closely correlated to the progression of GC. Afterward, the prognostic risk score was constructed, and values were verified. Patients in the high-risk group revealed a worse prognosis, which was positively correlated with the VIM gene and negatively correlated with the CDH1 gene. Of note, CHRNB2 expression was significantly suppressed only in the TP53 mutation group of GC patients [25]. Also, a study identifying methylated-differentially expressed genes (MDEGs) on a genome-wide scale systematically explored Multi-Omics cohorts from the TGCA and GEO to construct a new MDEGs-based signature [26]. The eight-MDEGs signature (TREM2, MICAL2, INHBA, PCSK5, NRP1, YAP1, RAI14, and MATN3) was identified and validated the prognosis performance. The multivariate Cox regression analysis presented that the eight-MDEGs signature risk score showed statistical significance as an independent variable in the TCGA-GC and GSE 84437. These MDEGs significantly divide GC patients into high- and low-risk groups in OS.

The DNA methylation patterns may have clinical implications as prognostic markers of GC patients, which provide information useful for detecting therapeutic targets. DNA methylation investigated as potential prognostic GC markers are listed in Table 2.

## 4. Transcriptmics-Based Validation of Predictive and Prognostic Markers of GC

A transcriptomic investigation analyzes an organism’s whole RNA content. Transcription is the method of copying a segment of DNA into RNA and reveals an outline of the activity of the cells [27]. Recently, transcriptomics has remarkably progressed in molecular genetics [28]. Transcriptomes denote all RNA molecules created in a cell from the genome under certain pathological situations [29,30]. Recently, the expression profile of G-protein-coupled receptor 27 (GPR27) in GC was investigated based on TCGA and GEO database [31]. GPR27 mRNA levels were lower in GC tissues than in normal gastric specimens. Missense mutation of GPR27 was found in 4% of GC patients through cBioPortal, and GPR27 mutation was associated with poor prognosis and GPR27 cg03024619 showed the most prognostic significance. Interestingly, GPR27 had an apparent interaction with the activation of T cells and tumor mutation burden.

Although bulk RNA-seq analyzes only the mean transcript expression in a cell population, single-cell RNA sequencing (scRNA-seq) evaluates the transcriptomic condition of specific populations of single cells [32]. A study using an immune-targeted single-cell profiling strategy examined the role of tumor immune microenvironment (TIME) in the progression of gastric signet ring cell carcinoma (GSRCC) [33]. TIME heterogeneity between GSRCC and non-GSRCC was compared by scRNA-seq. They discovered that the GSRCC TIME may be dormant, where Treg-FOXP3 and CD8-Tex were hard to mobilize and impaired B cells’ accurate activities. They also indicated the potential role of CXCL13 in forming the TIME within GSRCC, suggesting that the diminution of CXCL13-mediated regulatory effect in CD8-Tex would be associated with the poor prognosis of GSRCC which is validated by the cytometry by time of flight (CyTOF) results. In another study, RNA-seq-based transcriptomics and TMT liquid chromatography-mass spectrometry (LC-MS)/MS-based proteomics study were conducted in which 14 differentiating genes were identified. These 14 genes were significantly enriched in metabolic pathways, suggesting that metabolic genes have a pivotal role in GC [34]. They also found that the gene set was negatively associated with Treg, monocytes, Th1, DC, and macrophages and positively correlated to NKT, CD4^+^, and MAIT cells. Besides, some metabolic genes such as ADH1B, PHGDH, BCAT2, NE3, PCCB, and CS were positively associated with PD-1.

Non-coding RNAs (ncRNAs), which comprise more than 90% of the RNA transcripts of the human genome, are deficient in protein-coding activity. However, accumulating evidence has found that ncRNAs contribute to the initiation and progression of GC and can act as important biomarkers in prognosis [35,36]. Hence, detecting differentially expressed ncRNAs may be associated with the prediction of survival in GC. A recent study using the gene expression profiles of the GEO database of GC identified 110 consistent differentially expressed miRNA (DEMs) between GC and paired normal tissues, which were processed with the limma package in R [37]. Subsequently, to validate the relationship between DEMs and the prognosis of GC patients. 361 specimens from TGCA were collected. Three DEMs (miR-145-3p, miR-125b-5p, and miR-99a-5p) were associated with poorer prognosis by using a log-rank test and Kaplan-Meier curve. The multivariate analysis showed that the T stage and the three-DEM signature were all independent factors in predicting the prognosis of patients with GC [37]. Another study examining intestinal GC molecular networks’ gene-microRNA relationship has shown that three regulatory circuits are identified in a complex interaction network [38]. Besides, enrichment analysis revealed that KRAS, PI3K, and p53 were associated with these regulatory circuit networks. Noteworthy, some of the hub components operated together and contributed to prognosis. The upregulation of hsa-mir-200b and the downregulation of its target *CFL2* were closely correlated with worse survival.

Growing evidence has shown that long non-coding RNAs (lncRNAs), with a length of more than 200 nucleotides, can manage cancer development and the alterations of variable genes [39]. Aberrant expression of several lncRNAs is involved in the pathogenesis, suggesting that novel lncRNAs may serve as the new clinical biomarkers. A study using a variety of machine learning algorithms and constructed immune-related lncRNA prognostic model (ILPM) detected 18 prognostic immune-related lncRNAs (SNHG5, LINC01270, CHKB. AS1, NUTM2A.AS1, MIR181A2HG, CCNT2.AS1, DLG3.AS1, LINC01134, NIFK.AS1, RP11.443B7.1, LSAMP.AS1, HMGN3.AS1, LPP.AS2, RP11.710C12.1, RP11.155O18.6, CASC15, RP11.449P15.2, and FLJ16779) [40]. All patients were classified into high- and low-risk groups depending on the median risk score. Patients belonging to the high-risk group showed poor outcomes, lower immune cell infiltration, and a higher metabolism. Notably, 86 DEGs were mainly enriched in immune-related pathways between high- and low-risk groups, and 18 drugs in the Genomics of Drug Sensitivity in Cancer were sensitive to the high-risk patients. LncRNAs also have been shown to promote the evasion of immune surveillance by cancer through various mechanisms. The competitive endogenous RNAs (ceRNA) hypothesis declares that as ceRNAs, some RNAs can mediate the expression of downstream mRNA by linking distributed miRNAs [41]. The integrative analysis of GEO and TCGA data reported that three lncRNAs (UCA1, HOTTIP, and HMGA1P4) may be involved in the development, and their prospective activities may be related to the prognosis of GC [42]. CeRNA network analysis found an interaction between hsa-miR-508-3p and HMGA1P4. Enrichment analysis showed that genes are enriched in the cell cycle and cellular senescence. Similarly, seven glycolysis-related lncRNAs (AL353804.1, AC010719.1, TNFRSF10A-AS1, AC005586.1, AL355574.1, AC009948.1, and AL161785.1) were identified based on the TCGA-STAD databases using LASSO and Cox regression analyses. These lncRNAs were enriched in the cell adhesion molecules, leukocyte transendothelial migrations, calcium signaling pathway, and JAK-STAT signaling pathway [43]. Studies using transcriptome in the prediction/prognosis of GC in the past 5 years are collected in Table 3.

## 5. Proteomics-Based Validation of Predictive and Prognostic Markers of GC

Proteins are vigorous regulatory elements of cells that sustain cellular processes. The normal activities of proteins may be disordered due to stress, which results in abnormal cell activity. Proteomics analysis can reflect the alterations in the final regulatory products of the cells. Recent technologies allow to prediction of protein expression change, which may mediate the cellular biological processes [46]. Proteomics study ordinarily examines the construction and functions of proteins, and activities during carcinogenesis. Proteomics biomarkers are generally protein components derived from the body’s system between normal and pathological conditions. The developments in knowledge and technologies of high-throughput mass spectroscopy (MS) provoke the identification of effective biomarkers for GC [47]. In this section, we aim to provide recent developments in finding biomarker candidates and the prospective application of MS-based biomarkers for detecting highly specific and sensitive prognostic biomarkers with their outlooks in clinical practice (Table 4).

A recent study established a prognostic risk model [48]. Three proteins (COLLAGEN VI, CD20, TIGAR) were identified and may be utilized as an independent factor that is closely associated with GC prognosis. Enrichment analyses showed that DNA metabolic and growth regulation pathways were enriched. Likewise, a study characterized by RPPA-based functional proteomic data identified the functional proteome signatures (*MYH11*, *CD20*, *CHK1_pS345*, *AR*, PR, *HER3*, *MYH11*, and *SMAD1*), suggesting that these candidate protein markers can serve as potential prognostic biomarkers in GC [49]. Another proteomic analysis attempted to characterize molecular changes in the extracellular matrix (ECM) surrounding GC. ECM components of tumor tissues revealed no difference from normal mucosa, whereas their levels were generally displayed as increased ECM proteins (LOX, LAMB3, ADAMTSL1, and FBN2) and decreased basement membrane components (COL4A5, COL4A6, COL28A1, NID1, and LAMA5) that were closely related to tumor angiogenesis, invasion, and metastasis, which were closely related to OS of GC patients [50]. Integrating proteomics with nano-liquid LC-MS identified a total of 429 differentially expressed proteins [51]. Enriched analysis revealed that several pathways were activated and correlated with chaperone function, apoptosis, and immune response. Generally involved pathways are the overexpression of heat shock protein (HSP) families, which are commonly expressed in response to stress, the antiapoptotic pathway, the BH3 interacting domain, and the BLC-protein family. Trefoil factor 1 and 2, chromogranin A, REGIA, and the mitogen factor gastrin 7 were downregulated. In addition, among the 20 hub proteins detected, IST1, VAT1, ALYREF, and HEXB were closely related to worse survival [51]. Interestingly, a recent study applied a deep learning-based proteomic subtyping workflow that inserted the autoencoder framework. Patients with a good prognosis showed effective responses to chemotherapy, whereas patients with a poor prognosis resisted. Enrichment analysis showed patients with good revealed outcomes of the pathways that were related to DNA replication, cell cycle, and programmed cell death. Meanwhile, patients with poor outcomes are enriched in ECM-related pathways [52].

**Table 4 ijms-25-05880-t004:** Proteomics-based predictive/prognostic GC biomarkers using bioinformatics analysis for the past 5 years.

Number of Patients	Databases	Analytical Platform	Statistical Analysis	Proposed Proteomics-Based Biomarkers	Comments	Refs.
18	GEO EGA TCGA	Nano-LC-MS PLGS ExpressionE tool algorithm MetaCore^TM^ KEGG Mapper PPI analyses	Kaplan–Meier Method	HSP, CHGA, TFF 1 and 2, REGIA, MAPK1	IST1, VAT1, ALYREF, and HEXB were closely related with the worse survival.	[51]
48	TCGA TIMER GSCA	TMT-MS	Kaplan–Meier Plotter database	BCAT2, ALDH6A1, MCEE, PCCB, BCKDHB, DBT, AUH	The expression of BCAT2 was suppressed in cancerous samples from GC patients and was negatively correlated with survival.	[33]
352	TCGA TCPA RPPA	dplyr, ggplot2, and ggrepel software packages, https://ggrepel.slowkow.com. GO and KEGG analyses	Kaplan–Meier survival curve Cox regression analysis	COLLAGEN VI, CD20, TIGAR	Tri-protein was detected as an independent prognostic factor.	[48]
N.A.	TCGA TCPA RPPA	MIFS algorithms PCA, PLS-DA, t-SN, and heatmap analysis	Kaplan–Meier survival analysis Cox regression analysis	MYH11, CD20, CHK1_pS345, AR, *PR*, HER3, MYH11, SMAD1	RPPA-based functional proteomic data identified the functional proteome signatures.	[49]
367	TCGA-STAD	Nano LC-MS/MS Proteome Discoverer 2.4.0.305 software InterPro Protein Domains and Features KEGG pathways analysis mRNA-Seq	Student’s *t*-test or One-way ANOVA Kolmogorov–Smirnov test Brown–Forsythe test	LOX, LTBP2, and COL1A2 provide disease progression and patient poor prognosis.	ECM constituents that characterize the matrisome of GC contexts were identified.	[50]

Abbreviations: GC, gastric cancer; TCGA, The Cancer Genome Atlas; GEO, Gene Expression Omnibus; EGA, European genome-phenome archive; MS, mass spectrometry; KEGG, The Kyoto Encyclopedia of Genes and Genomes; PPI, protein–protein interaction; HSP, heat shock protein; TFF, trefoil factor; MAPK, mitogen-activated protein kinase; CHGA, Chromogranin A; TMT-MS, tag-mass spectrometry; TCPA, The Cancer Proteome Atlas; RPPA, reverse-phase protein array; PCA, principal component analysis; PLS-DA, partial least squares discriminant analysis; N.A., not applicable.

## 6. Metabolomics-Based Validation of Predictive and Prognostic Markers of GC

Metabolomics is a comparatively novel field of study that measures all the metabolites within a biological sample (metabolome) and the purposes of quantifying metabolites in a metabolome [53]. Metabolomics can amplify small alterations in the genome, transcriptome, and proteome, which indicate the termination of gene activities and changes in individuals [54]. Thus, metabolomics is thought to be the last direction of omics research. Unlike normal cells, cancerous cells have been shown to present an extremely marked metabolic phenotype, which is depicted by amplified energy synthesis, elevated antioxidant regeneration, and enhanced macromolecule production. Commonly, cancer metabolism has a significant role in the tumorigenesis, progression, as well as drug resistance of cancer [55]. Compared with other “omics” involving genomics, transcriptomics, and proteomics, metabolomics could provide direct and general information on metabolites and result in response to both genetic alterations and pathological stimulation [56]. Over the past few years, with the advancement of metabolomics technology, previous studies have examined the application of metabolomics to identify novel biomarkers and uncover prediction biomarkers for prognosis in GC [57] (Table 5). Aberrant metabolites in metabolism pathways, such as carbohydrate, lipid, amino acid, and nucleotide metabolisms, may be useful for developing potential biomarkers for GC prognosis and prediction for GC therapy. In addition, metabolomic analysis is cost-effective compared with a genome, a transcriptome, or a proteome analysis.

A study screening GC survival-associated metabolites identified 24 metabolites by using Kaplan–Meier survival analysis [58]. Eight metabolites (ornithine, phenyl acetyl-L-glutamine, porphobilinogen, linoleic acid, DL-dipalmitoyl phosphatidylcholine, 5′-mdthylthioadenosine, inosine triphosphate, and paraxanthine) were indicated as an independent prognostic factor of GC by the forest plot of the Cox regression analysis. Enrichment analysis found that purine metabolism and linoleic acid metabolism were found to be the most significant metabolic pathways related to the survival of patients with GC. Another metabolomics study proposed to detect differential levels of metabolites in the pleural effusion of GC patients. They divided the patients into two groups depending on the existence of peritoneal metastasis, serosal invasion, and CEA mRNA [59]. Seventeen differential metabolites (sulfite, TG (54:2), TG (53:4), G3P, α-aminobutyric acid, α-CEHC, dodecanol, glutamyl alanine, 3-methyl alanine, 3-hydroxysteroid, CL (63:4), PE-NMe (40:5), retinol, MG (21:0/0:0/0:0), tetradecanoic acid, tridecanoic acid, octacosanoic acid and myristate glycine) were found with a weight of 100% using SVM. These metabolites had potential predictive ability and independent risk factors for peritoneal metastasis of GC patients.

Lipid metabolism deregulation is one of the most familiar metabolic alterations in cancer [60]. Interestingly, a study examining differential serum metabolites showed significant alterations in the abundance of several metabolites among patients before and after surgery [61]. Particularly, lysophosphatidic acids, triglycerides, lysine, and sphingosine-1-phosphate revealed increased levels in the postoperative groups compared with the preoperative and recurrence groups. Meanwhile, phosphatidylcholine, oxidized ceramide, and phosphatidylglycerol displayed decreased levels. These results may suggest that these differential metabolites may be useful biomarkers for validating the prognosis and monitoring the recurrence of GC. A more recent study that employed LC–MS-based targeted metabolomics to illustrate the global metabolomic profiling of large cohorts analyzed plasma samples from GC patients utilizing machine learning [62]. Machine learning-derived 28-metabolite prognostic model showed successful predictive activity with an AUROC of 0.832 and a concordance index of 0.83. Among them, 11 of the 28 metabolites (symmetric dimethylarginine/asymmetric dimethylarginine, neopterin, thymine, glucuronate, hydroxyproline, 14:0 Carnitine, indole acrylate, 8:0 Carnitine, acetylalanine, 2-aminoadipate, and GlcNAc6p) significantly discriminated the prognosis of GC patients. Patients who are distinguished by the 28-PM model from the high-risk group are more likely to reveal worse survival compared with the low-risk group and provide an advantage from intensive monitoring, intervention, and clinical trial. A study by Song and colleagues attempted to detect metabolic features for gastrointestinal cancer prognostic assessment [63]. MS-based platforms analyzed plasma samples, and 133 metabolites were significantly altered in the GC samples. Pathway enrichment analysis displayed that they are principally involved in steroid hormone biosynthesis, cysteine and methionine metabolism, arginine and proline metabolism, galactose metabolism, amino acids, and purine metabolism pathways. Ubiquinone and other terpenoid–quinone biosyntheses, phenylalanine, tyrosine, and tryptophan biosynthesis, histidine metabolism, starch and sucrose metabolism, fructose and mannose metabolism, and inositol phosphate metabolism were associated with GC patients. Interestingly, metabolic patterns changed before- and after surgical procedures in GC. The levels of PE 38:4, threonic acid, l-lysyl-l-glutamine, and temorine were relatively high after GC surgery.

Genome-wide association studies (GWAS) can identify hundreds of quantitative trait loci vigorously related to plasma/serum levels of metabolites. Studies integrating metabolomics and genetics analyses, known as metabolite GWAS (mGWAS), enable the examination of correlations between a variety of gene alterations and metabolites and a valuable understanding of the genetic control of metabolite levels [64]. A recent study investigated the genetic variation mechanism of GC prognosis-related metabolites using mGWAS. It detected 32 genetic variant loci that were potentially relevant to GC survival-related metabolites, corresponding to 7 genes, including *VENTX*, *PCDH7*, *JAKMIP1*, *MIR202HG*, *MIR378D1*, *LINC02472*, and *LINC02310* [65]. In addition, gene oncology (GO) enrichment analysis found that these genes are principally associated with cell signaling molecular transmission, organism growth and development, nervous system regulation, and immune escape-related pathways, suggesting that these functional genes regulate the metabolites via related pathways and thereby affect GC survival.

**Table 5 ijms-25-05880-t005:** Metabolomics-based predictive/prognostic GC biomarkers using bioinformatics analysis for the past 5 years.

Number of Patients	Analytical Platform	Statistical Analysis	Proposed Proteomics-Based Biomarkers	Comments	Refs.
218	Agilent MassHunter qualitative analysis software (version B.01.00, Agilent Technologies, Santa Clara, CA, USA) MetaboAnalyst	Kaplan–Meier analysis log-rank tests Multivariate Cox regression analyses Lasso regression	omithine, phenylacetyl-L-glutamine, porphobilinogen, linoleic acid, DL-dipalmitoyl phosphatidylcholine, 5′-methylthioadenosine, inosine triphosphate, paraxanthine	These metabolites were indicated as an independent prognostic factor.	[58]
65	LC-MS Total ion current spectra PCA	*t*-test, Support vector machine	sulfite, TG (54:2), TG (53:4), G3P, α-aminobutyric acid, α-CEHC, dodecanol, glutamyl alanine, 3-methyl alanine, 3-hydroxysteroid, CL (63:4), PE-NMe (40:5), retinol, MG (21:0/0:0/0:0), tetradecanoic acid, tridecanoic acid, octacosanoic acid and myristate glycine	These metabolites had independent risk factors for peritoneal metastasis of GC patients.	[59]
31	LC-MS	*t*-test	lysophosphatidic acids, triglycerides, lysine, and sphingosine-1-phosphate, phosphatidylcholine, oxidized ceramide, phosphatidylglycerol	These differential metabolites may be useful biomarkers for validating the prognosis and monitoring the recurrence of GC.	[61]
181	LC-MS Random survival forest method LC-MS analysis Wilcoxon rank-sum test KEGG pathway enrichment analysis	Kaplan–Meier curves LASSO regression algorithm	symmetric dimethylarginine/asymmetric dimethylarginine, neopterin, thymine, glucuronate, hydroxyproline, 14:0 Carnitine, indole acrylate, 8:0 Carnitine, acetylalanine, 2-aminoadipate, and GlcNAc6p	Eleven metabolites significantly discriminated the prognosis of GC patients.	[62]
37	LC-MS GC-MS Waters ACQUITY UPLC ChromaTOF TargetSearch	Normalization Autoencoder *t*-test	PE 38:4, threonic acid, l-lysyl-l-glutamine, temorine	These metabolites differed before and after surgery.	[63]
218	mGWAS GO enrichment analysis Cytoscape software, https://cytoscape.org/	Multifactorial Cox regression analysis	*VENTX*, *PCDH7*, *JAKMIP1*, *MIR202HG*, *MIR378D1*, *LINC02472*, and *LINC02310*	These functional genes regulate the metabolites via related pathways and thereby affect GC survival.	[65]

Abbreviations: GC, gastric cancer; LASSO, least absolute shrinkage and selection operator; LC-MS, liquid chromatography-mass spectrometry; PCA, principal component analysis; GC-MS, gas chromatography-mass spectrometry; KEGG, The Kyoto encyclopedia of genes and genomes; UPLC, ultra-performance liquid chromatography; mGWAS, metabolite genome-wide association studies; N.A. = not applicable.

These newly detected metabolites may be significant prognostic markers or predict therapeutic targets for patients with GC.

## 7. Pharmacogenomics-Based Validation of Predictive Markers of GC

The prediction of treatment efficiency, such as chemosensitivity, is a major challenge in the management of GC. To select efficient drugs, recent clinical trials have focused on multi-omics profiling of patients with cancer by analyzing genome-wide data, including data from DNA sequencing, gene expression profiling, and copy number profiling [66]. Pharmacogenomics proposed to understand how genetic variants affect drug response and toxicity. The capacity to predict the efficacy of a patient with cancer on a particular treatment regimen is the elaborate goal of precision oncology [67].

The large-scale pharmacogenetic study using a drug sensitivity dataset based on patient-derived GC specimens identified the lineage-specific drug sensitivities, including the effect of epidermal growth factor receptor inhibitors for signet-ring cells, vascular endothelial growth factor receptor inhibitors for diffuse-type tumors, PARP inhibitors for tumors with BRCA2 mutation, WNT inhibitors for ALK-mutant tumors, and AKT blockade for PIK3CA-E542K mutation [68]. A study proposed to build a predictive model to guess the susceptibility to platinum of GC patients to excuse patients from unnecessary exposure to toxicity [69]. Twenty-three genes that are markedly relevant to the survival of patients with GC were detected by using univariate Cox regression analysis. Subsequently, three potential genes were selected to establish the predictive model consisting of ND6, BRMS1, and SRXN1. This model can accurately predict the outcome of GC patients after platinum-based treatment. Based on in silico data mining, the treatment benefit of up to 15% of GC samples from the TCGA dataset can be predicted by currently approved targeted therapy for GC (trastuzumab, pembrolizumab, and ramucirumab). In contrast, 50% of the patients would be predicted to benefit from FDA-approved alternative targeted therapies. A major GC target identified in this analysis was the high frequency of PIK3CA mutation variants. Other drugs identified here for therapeutic options are sorafenib (16% of TCGA GC patients), cobozantinib, and regorafenib [70]. Another study has described that six lactylation-related prognostic gene models in GC tissues were established using the GSEA, TCGA, and GEO databases [71]. Lactylation score was achieved by infiltrated immune cells and genetic instability amounts, and it was shown that lactylation score was intensely associated with GC prognosis. GC tissues with a high lactate fraction have higher immune dysfunction and reduced response to immune checkpoint inhibitors, indicating higher risks of immune evasion and dysfunction. These findings suggested that the lactate score may be useful for predicting prognosis and immune escape of GC. A recent study investigated the indicators for the efficacy of first-line chemotherapeutic agents, such as XELOX (capecitabine and oxaliplatin), DOS (docetaxel, oxaliplatin, and S-1), and targeted therapies, such as anti-HER2-based therapy by employing comprehensive proteomic technology implemented with machine learning statistics of 206 tumor samples before treatment [72]. The proteomic subtypes were established and correlated with therapy subcohorts. HER2 subcohort showed distinct distribution differences among proteomic subtypes. The expression of one of the ECM proteins, THSD4, was closely relevant to worse survival and associated with chemosensitivity (involving 5-FU, oxaliplatin, and docetaxel) in GC cell lines. Besides, highly expressed extracellular matrix proteins (THSD4, SRPX2, TGFBI, THBS1, and LAMB2) can serve as indicators to predict drug resistance. They also found PI3K-AKT signaling pathway had potential relevance to the ECM proteins (COL4A1, COL6A5, FN1, GP1BA, ITGA4, THBS3, and THBS4), and downregulation of apoptosis correlated proteins (BCL2, BCL2L1, CASP3, CASP7, and CDKN2A), suggesting the synergistic efficacy of PI3K-AKT inhibitor combined with anti-HER2 targeted agents. Interestingly, increased expression of MSI-sig CTSE enhances the effectiveness of DOS therapy but not of XELOX therapy. Meanwhile, TKTL1 is negatively associated with the sensitivity of patients to DOS therapy.

Previous studies have indicated that metabolomics analysis in GC may offer a useful approach to chemosensitivity prediction [73,74,75]. Recently, a study examined the role of metabolite profiles in sensitivity to neoadjuvant chemotherapy in GC [76]. They collected serum samples from GC patients with diverse sensitivities to chemotherapy. LC-MS analysis found that four serum metabolites (Deoxyribose 1-phosphate, S-lactoylglutathione, lysoPC (16:0), and O-arachidonoyl ethanolamine) revealed potent divergences in patients with chemo-sensitive and chemo-resistant tumors with a sensitivity and specificity of 82.5%. Besides, deoxyribose 1-phosphate and S-lactoylglutathione were independently correlated with chemosensitivity by multivariate regression analysis. The approved biomarkers and candidate biomarkers for therapeutic prediction for targeting GC using bioinformatical analysis are summarized in Table 6.

The results obtained by network analysis, combined with multi-omics data via bioinformatics algorithms, strongly suggest that the selection of therapies for patients with GC should consider these multi-omics profiles for effective individualized treatment strategies.

## 8. Discussion and Future Directions

This review summarizes single omics and potential multi-omics applications for predictive and prognostic biomarkers in GC. Predictive biomarkers can be used broadly to predict the prognosis and therapeutic effects and identify individuals at high risk. In this review, we focus on the role of predictive biomarkers in predicting therapeutic strategies. For a more specific patient stratification along with prognosis and therapeutic responses, comprehensive profiling using integrated bioinformatics analysis enables the classification of GC cases by molecular subtypes. As mentioned above, numerous studies have analyzed the genetic, epigenetic, transcriptomic, proteomic, or metabolomic signatures of GC and found multiple molecular alterations associated with GC. The fusion of this evidence has prompted TCGA and the Asian Cancer Research Group (ACRG) to propose several molecular classifications [77,78]. However, rational application of these classifications in clinical settings is still not widespread, mainly due to their intricacy. Beyond these studies, many researchers have sought other surrogate molecular classifications of GC to detect biomarkers and validate therapeutic prediction based on various analyses, including epigenome, proteome, metabolome, immunity, and tumor mutational burden [56,72,79,80,81,82,83,84,85,86,87]. However, no studies can be said to have succeeded in identifying alternative markers that allow more efficient categories than those of TCGA and ACRG. The application of novel molecular techniques beyond NGS, such as third-generation and single-cell sequencing, and the classification of patients in clinical trials according to molecular alterations may eventually lead to a consensus molecular classification [88].

Reproducibility, clinical feasibility, and validation in randomized clinical trials would make biomarkers more clinically relevant. To translate research results into the clinical use of a biomarker for cancer, findings must be validated by extensive clinical trials across diverse institutions. Currently, there are several bioinformatics-based clinical trials for patients with GC: three trials are ongoing (NCT06035250, NCT05319392, NCT05985577), and one trial is completed (NCT05620537). In addition, a recently completed clinical trial examined nutritional status assessments, such as Nutrition Risk Screening 2002 score and Patient-generated Subjective Global Assessment in patients with GC after radical surgery; its analysis featured the LASSO regression model as well as Cox regression (ClinicalTrials.gov identifiers: NCT05620537). A multicentered prospective cohort study is evaluating the efficacy of neoadjuvant chemotherapy and prognosis in patients with GC using a deep-learning-based predictive model integrating patients’ clinical data, biopsy pathology images, and CT imaging data (NCT06035250). Patients with GC are currently being recruited for a clinical trial to search the transcriptome profiling of GC tumor samples and adjacent normal tissue to examine DEGs based on TCGA and GEO. DEGs between cancerous and normal tissue were detected and annotated using bioinformatic analysis to gain insight into the TME (NCT05319392). Similarly, a clinical trial characterizing the prognostic proteomic landscape of GSRCC using LC-MS/MS and clinical outcome data is currently underway (NCT05985577).

Although it has recently become easier to acquire multi-omics information that can involve several layers at once, it is often challenging to fully use the gathering of large amounts of omics data available, and integrating the omics into a significant model is difficult due to the number of datasets. Thus, various integration strategies have been developed. Recent advances in proteomics through the maturation of several MS techniques have enabled the introduction of proteogenomic approaches that can integrate genomic data with proteomics and information on post-translational modifications [89]. For instance, there has been deep attention to the comprehensive characterization of the divergent mechanisms of diffuse-GC and intestinal-GC [90]. For instance, integrative proteogenomic analysis showed that the *ARID1A* mutation conferred contradictory prognostic influences in the two different entities. Meanwhile, diverse effects on corresponding proteomes were demonstrated, implying that diffuse-GC and intestinal-GC should be characterized based on multilevel data, including proteogenomics. More recently, comprehensive proteomic profiling using LC-MS has been applied to peritoneal carcinomatosis cells from GCs and compared with transcriptomic profiles [91]. Unsupervised hierarchical clustering analysis revealed three subgroups based on the extent of enrichment in cancer cells: Cluster A (tumor-cell enriched), Cluster B (tumor-immune mixed), and Cluster C (immune-cell enriched), which may be correlated with prognosis and drug response. When mutational profiles were integrated into proteomic data, many somatic mutations in driver genes and a higher tumor mutation burden were detected in Cluster A, while no somatic mutations were found in Cluster C. Pathway enrichment analysis displayed several enriched biological pathways (glycolysis, oxidative phosphorylation, TP53, MYC, mTORC1, TGF-β, neutrophil degranulation, neutrophil activation, neutrophil-mediated immunity, and cytokine production pathways) and several druggable targets (cancer-testis antigens, kinases, and receptors). Proteogenomic analysis revealed several genes with conflicting protein and mRNAs expression in peritoneal carcinomatosis cells, such as CTNNA2, LTF, CTAGE1, LAIR1, and HAVCR2, related to oncogenic, metabolic, and immune-related pathways, suggesting that combining proteomic and transcriptional analysis might result in a better understanding of the complex mechanisms affecting protein expression in peritoneal carcinomatosis from GC [91]. Potentially actionable therapeutic molecular targets will be detected using proteogenomic approaches to study more about the molecular composition of patients and their tumors. It will then be possible to study the correlation between molecular discoveries and cancer treatment outcomes and to facilitate novel clinical trials with prognostic and predictive biomarkers.

Despite substantial advances in the detection of carcinogenesis, the association of cancer with genetic vulnerability to microbial infections remains a challenge. The microbiome has been associated with regulating various physiological processes in human diseases, including cancer, and this connection is attracting increasing attention [92]. Among all tumors, GC is arguably the most intensely affected by gut bacteria owing to their spatial proximity, and their associations with the gut microbiome have been intensively investigated [93,94]. As a standout example, *Helicobacter* pylori infection is one of GC’s most important risk factors [95]. A recent study characterized the mucosal microbiome composition in paired cancerous and adjacent normal tissues and found significant correlations with the prognosis of GC patients [96]. From 128 samples, 5450 different phylotypes belonging to 19 phyla and 296 genera were detected using a bioinformatics approach. The GC patients with enrichment of *Fusobacterium* nucleatum and the genus *Prevotella* I in their cancerous tissues showed worse survival using PERMANOVA and ANOSIM analyses. Bacterial network analysis revealed that F. nucleatum and *Prevotella* closely interacted with each other and with gastric bacterial communities in patients with GC, indicating that the gastric microbial environment may be responsible for the GC phenotype.

As described above, there are several obstacles to the extensive implementation of omics data in clinical practice. One problem is the necessity for large phenotypically characterized datasets. High accuracy and reproducibility are required for measurement technology in multi-omics studies. It is difficult to obtain the high-grade datasets needed for omics profiling in clinical practice due to the challenge of acquiring appropriate tissue samples, strict requirements for sample preparation, and high costs. Data mining may overcome these technical challenges. Omics data can be retrieved from multiple databases or collected with technologies for disease. These are often incompatible, which makes it difficult to compare the data obtained among different research groups. Normalization of patient characteristics with GC, such as age, race, and stage, from different databases, is very important to integrate multi-omics data. When it comes to the clinical utility of the presented biomarkers, they still need to be validated through comprehensive cohort studies and large-scale clinical trials. Further data standardization and the proceeding of dominant public databases for omics data will be required in the future. Each type of omics data is distinctive to a single “layer” of biological phenomena, and single-layer omics studies have offered a useful understanding of disease mechanisms. However, the interactions of the individual layers are complex and difficult to analyze. Recent studies have shown that integrative multi-omics analysis facilitates the identification of the networks that regulate transitions from health to cancer development and evaluation in greater depth of clinically related subgroups for potential biomarkers. Meanwhile, multi-omics data formats and storage methods are diverse. The majority of the multi-omics integrative analysis methods require specific formats for data. Thus, each omics dataset requires preprocessing, which comprises systematic normalization, filtering of data, and elimination of bunch effects (Figure 1). The data filtering step has a crucial role in reducing the noise and the volume of inputs.

Along with the introduction of high-throughput technologies and the availability of multi-omics data derived from a number of samples, a large set of multi-omics data integrative tools and methods have been reported for the discovery of biomarkers. These methods can be divided into categories, such as Bayesian integration (e.g., Pathway Recognition Algorithm using Data Integration on Genomic Models (PARADIGM) [97], iClusterPlus [98], Patient-specific data fusion (PSDF) [99]), network and pathway integration (e.g., Similarity network fusion (SNF) [100], PaintOmics [101], MultiGSEA [102], multi-omics gene-set analysis (MOGSA) [103], Cytoscape [104]), fusion-based integration (e.g., Protein fusion analysis (PFS) [105]), similarity-based integration (e.g., Perturbation clustering for data integration and disease subtyping(PINSPlus) [106], Neighborhood-based multi-omics clustering (NEMO) [107]), and other multivariate approaches (e.g., mixOMICS [108], moCluster [109], Integrative nonnegative matrix factorization (NMF) [110]) [111,112]. Considering such a broad spectrum of tools/methods to integrate multi-omics datasets, a detailed comparison and benchmarking of clustering methods utilizing identical datasets may be beneficial. In addition, there is a varied array of platforms that facilitate visualization and clarification of multi-omics data, involving GENEASE [113], CGDV [114], and SLIDE [115], which provide ease of visualization and interpretation of large biological data sets. Since the methods are widely available, the development of an identical framework can efficiently process and analyze multi-omics data with easy and unified analysis. Further comprehensive reviews of statistical and algorithmic aspects of integrative multi-omics data are accessible elsewhere [111,112]. Competitive endogenous RNAs (ceRNA) is a novel regulatory mechanism that has been found to be involved in a variety of progression processes in cancer [116]. Integrative multi-omics strategies using Cytoscape were applied to examine the relationship between the prognosis and biomarkers in the STAD-linked ceRNA network by Kaplan–Meier survival analysis [45]. Fifteen genes were associated with the outcome of GC patients, involving AC011374.1, AC018781.1, ADAMTS9-AS1, AL139002.1, AL391152.1, HOTTIP, FLRT1, NKX2-1-AS1, ADAMTS9-AS2, LINC00326, VCAN-AS1, SERPINE1, POU6F2-AS2, IGF2-AS, and miR-145. Among them, in the ceRNA network, the lncRNAs VCAN-AS1, AL139002.1, LINC00326, AC018781.1, and C15orf54 acted as hub nodes. In particular, this study also found an association between TIME, (including monocytes, and neutrophils) and lncRNA-based ceRNA regulatory network [45]. In recent years, with the rapid progress of artificial intelligence, deep learning has been continuously developing. Interestingly, Chen and colleagues demonstrated a deep learning-based model on GC to find multi-omics features that were associated with the distinctive survival of GC patients [117]. They preprocessed the data and acquired 16,699 genes from RNA-seq, 390 miRNAs from miRNA-seq, and 18,992 genes from DNA methylation data as input features. These three types of omics features were stacked together using an autoencoder deep learning architecture. By the silhouette index and the Calinski–Harabasz criterion, two subgroups were identified from k-means clustering, and a significant difference in prognosis was shown between the two subgroups. Notably, implementation of the model based on the autoencoder was superior to those of two alternative familiar integrative multi-omics tools, principal component analysis and similarity network fusion. Membership in one subgroup was deemed to be a risk factor using the univariate and multivariate Cox-PH model, and the model based on the autoencoder-integrating multi-omics analysis showed more suitable performance as a prognostic marker than other clinicopathological features [117]. Thus, proper omics data creation and management and the selection of data integration methods under the complementary computational analysis, could be essential for providing the effective application of multi-omics datasets in clinical translation.

Collectively, the practical integration of bioinformatics analysis and multi-omics datasets is having a great impact on the search for predictive and prognostic biomarkers and may lead to an important revolution in treatment.

## 9. Materials and Methods

A non-systematic review was performed based on an electronic search through the medical literature using PubMed and Google Scholar. The keywords “gastric cancer”, “bioinformatics analysis”, “multi-omics”, “artificial intelligence”, “machine learning”, “prognosis”, “prediction”, and “biomarker” were used. Review articles and guidelines (from 2018 to 2023) investigating the current status of bioinformatic approaches in understanding GC’s genetic, epigenetic, transcriptomic, proteomic, and metabolic processes were included in this review. When more than one guideline concerning the same subject was available, the most recently updated guideline was selected. Only full articles published in English within the last ten years were considered for further review. Close attention was also given to “clinical study” and “review” articles dealing with the topic. The exclusion criteria were duplicate articles and studies lacking diagnostic outcomes. Case reports, correspondence, letters, and non-human research were excluded.

## 10. Conclusions

While not fully realized, the era of comprehensive molecular analysis, including not only genomics but also epigenomics, transcriptomics, proteomics, and metabolomics data, has arrived. In this review, we have summarized the use of these omics layers in the detection of predictive and prognostic biomarkers in GC. It is expected that the use of bioinformatics analysis to identify potential biomarkers leads to better therapeutic strategies. However, further elucidation is needed to apply these biomarker candidates in a clinical setting. The practical, integrative analyses of multi-platform datasets via machine learning could assist in validating novel and specific biomarkers associated with prognosis and prediction and may provide opportunities for clinical translation in the future.

## Figures and Tables

**Figure 1 ijms-25-05880-f001:**
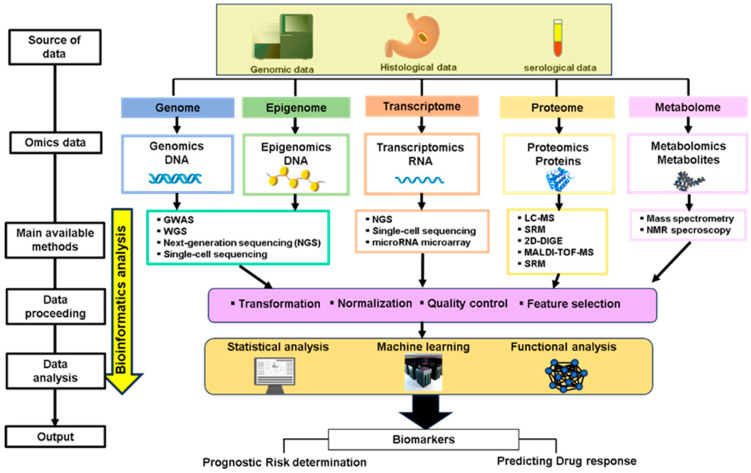
Schematic diagram of validation of biomarker by the integration of bioinformatic analysis with multi-omics platforms. Different omics technologies and their features, such as omics targeted, sequencing techniques, data processing, data analysis, and output, were displayed. The first step is the identification of high-quality samples obtained by blood, tissue samples from GC patients, or high-throughput datasets. Multi-platform omics technologies are analyzed individually to identify the DNA, RNA, proteins, and metabolites responsible for the GC progression. Omics data are extracted by the search for various omics databases, mining of the data, and obtaining the pre-processed omics data. Thereby, data are transformed and processed through normalization, quality control, and feature selection to mine interpretable information. These multidimensional data can be integrated based on statistical analysis, machine learning, and functional analysis, which lead to the discovery of putative biomarkers.

**Table 1 ijms-25-05880-t001:** Genomics-based predictive/prognostic GC biomarkers using bioinformatics analysis for the past 5 years.

Number of Patients	Databases	Analytical Platform	Statistical Analysis	Gene set Identified	Comments	Refs.
415	TCGA	EdgeR and DESeq packages GO and KEGG pathway enrichment analyses	Cox regression analysis	*PCOLCE2*, *TMEM132C*, *UPK1B*, *SLITRK2*, *PM20D1*, *FLJ3502*	Six distant metastasis-related genes can be an independent indicator of the prognosis of GC and as a clinically useful classification tool to facilitate individualized GC therapy.	[13]
N.A.	GEO (GSE84433, GSE26942) TCGA	Limma and pheatmap package WGCNA GO and KEGG pathway enrichment analyses	Kaplan–Meier Plotter	*CST1*, *CEMIP*, *COL8A1*, *PMEPA1*, *MSLN*	Five genes were identified as the hub genes by utilizing WGCNA. *CEMIP* plays an important role in GC progression	[16]
375	TGCA TIMER	Limma package WGCNA	Cox analysis	*RXRG*, *AGT*, *BCHE*, *UBE2QL1*, *PLCXD3*, *ADCYAP1R1*, *NRCAM*, *MAMDC2*, *CDH19*, *GAMT*	AGT expression is higher in GC and *BRD9*, *GOLPH3*, *NOM1*, *KLHL25*, and *PSMD11* interact with *AGT*, which is closely related to the poor prognosis.	[14]
60	GEO (GSE15459) TCGA	GO cellular component analysis	Cox regression analysis	*AGT*, *SERPINH1*, *MMP7*	The exosome levels of AGT and MMP7 were associated with the serum level and stimulated migration of GC cell lines.	[18]
747	GEO (GSE65801, GSE54129, GSE118916, GSE15459, GSE51575, GSE65801) TCGA-STAD	Limma and edgeR R packages GO and KEGG pathway enrichment analyses WGCNA PPI network TIMER	Kaplan–Meier survival analysis log-rank test	*TRYOBP*, *C1QB*	*TRYOBP* and *C1QB* were correlated with the CD8^+^ T cells, CD4^+^ T cells, and macrophages	[17]
351	GEO (GSE103236,GSE13911, GSE79973)	Geo2R analysis. DAVID, GO, and KEGG pathway enrichment analyses PPI network TIMER	Kaplan–Meier plotter	*CTHRC1*, *BGN*, *FAP*, *THBS2*, *COL12A1*, *COL5A2*, *SULF1*, *SPP1*, *COL10A1*	Nine genes negatively correlated with poor outcomes and an immune infiltrate based especially on immunosuppressive M2 macrophages.	[19]

Abbreviations: GC, gastric cancer; TCGA, The Cancer Genome Atlas; GEO, Gene Expression Omnibus; GO, gene oncology; KEGG, The Kyoto Encyclopedia of Genes and Genomes; WGCNA, weighted correlation network analysis; TIMER, Tumor Immune Estimation Resource; PPI, protein–protein interaction; N.A., not applicable.

**Table 2 ijms-25-05880-t002:** Epigenomics-based predictive/prognostic GC biomarkers using bioinformatics analysis for the past 5 years.

Number of Patients	Databases	Analytical Platform	Statistical Analysis	Gene set Identified	Comments	Refs.
343	GEO (GSE62254) TCGA cBioPortal GSEA	DESeq package DAVID ConsensusPathDB MethylMix analysis GO and KEGG pathway enrichment analyses	Kaplan–Meier analysis LASSO Cox regression model	*PODN*, *MYO1A*, *NPY*, *MICU3*, *TUBB6*, *RHOJ*	Six DNA methylation-driven DEGs identified using an OS nomogram were associated with good prognosis.	[24]
192	GEO (GSE13911, GSE30601, GSE79973, GSE25869, GSE15459) TCGA	Illumina microarray platform Geo2R analysis. DAVID, GO and KEGG pathway enrichment analyses	Cox and LASSO regression analysis Kaplan–Meier method	*TREM2*, *MICAL2*, *INHBA*, *PCSK5*, *NRP1*, *YAP1*, *RAI14*, *MATN3*	The eight-MDEGs signature risk score showed statistical significance as an independent variable.	[26]
443	GEO (GSE30601) TCGA	Wilcoxon test Pheatmappackage cBioPortal KEGG pathway enrichment analyses PPI network	Cox and LASSO regression analysis Kaplan–Meier method	*CEP290*, *CCDC69*, *UBXN8*, *KDM4A*, *AKR1B1*, *RASSF2*, *CHRNB2*, *EGR1*, *ARMC9*, *RPN1*	Eleven prognostic-related DMSs were positively correlated with the VIM gene and negatively correlated with the CDH1 gene.	[25]

Abbreviations: GC, gastric cancer; TCGA, The Cancer Genome Atlas; GEO, Gene Expression Omnibus; GO, gene oncology; KEGG, The Kyoto Encyclopedia of Genes and Genomes; PPI, protein-protein interaction; DMS, differential methylation site.

**Table 3 ijms-25-05880-t003:** Transcriptmics-based predictive/prognostic GC biomarkers using bioinformatics analysis for the past 5 years.

Number of Patients	Databases	Analytical Platform	Statistical Analysis	Gene Set Identified	Comments	Refs.
32	GEO (GSE212212)	scRNA-seq CellChat R package	T-test, Wilcoxon rank-sum test, and Fisher’s exact test.	*CXCL13*	GSRCC TIME may be dormant, where Treg-FOXP3 and CD8-Tex were hard to mobilize and impaired B cells’ accurate activities.	[33]
48	TCGA TIMER GSEA	RNA sequencing GO and KEGG pathway enrichment analyses	Kaplan–Meier Plotter database	*BCAT2*, *ALDH1A2*, *MDH1*, *PHGDH*, *CKB*, *ADH1B*, *PCCB*, *NNT*, *CKM*, *DCXR*, *LIPF*, *ASS1*, *ME3*, *CS*	Fourteen metabolic genes were significantly enriched and were correlated with PD-1.	[34]
361	GEO (GSE883415) TCGA	Limma package Functional Enrichment analysis	Kaplan–Meier method, Log-rank test Cox regression analysis	miR-145-3p, miR-125b-5p, and miR-99a-5p	The multivariate analysis showed that the three DEM signatures were all independent factors in predicting the prognosis.	[37]
180	TCGA	DESeq2 package MultiMiR package LC-MS/MS Enrichr	Kaplan–Meier method Log-rank test	hsa-mir-200b *CFL2*	These regulatory circuit networks were associated with survival.	[38]
97	GEO TCGA	GEPIA 2 TIMER cBioPortal GSEA	MethSurv Kaplan–Meier plotter	GPR27	GPR27 was useful for predicting prognosis and had a clear interaction with immune cells’ infiltration as well as their markers in patients with GC.	[34]
441	GEO (GPL6947, GSE84437) TCGA-STAD	RNA-seq DAVID, GO, and KEGG pathway enrichment analyses PPI network	Cox regression models	*RIMS1*, *PRICKLE1*, *MCC*, *DCLK1*, *FLRT2*, *SLCO2A1*, *CDO1*, *GHR*, *CD109*, *SELP*, *UPK1B*, *CD36*	Thirteen mRNA-based risk score model implemented acceptably in discriminating the risk of GC prognosis.	[44]
N.A.	GEO (GSE57303, GSE62254) TCGA-STAD GDSC	ImmLnc GSEA, GSVA The CIBERSORT algorithm GO and KEGG pathway enrichment analyses	Kaplan–Meier survival analysis Cox regression analysis Immune-related lncRNA prognostic model LASSO	SNHG5, LINC01270, CHKB. AS1, NUTM2A.AS1, MIR181A2HG, CCNT2.AS1, DLG3.AS1, LINC01134,etc	Among the 18 lncRNAs, the *p*-value of MIR181A2HG is the lowest, implying that it possesses good prediction activity.	[40].
N.A.	TGCA-STAD Ensembl CIBERSORT	Illumina HiSeqRNASeq platforms EdgeR analysis DESeq2 analysis Cytoscape GO and KEGG pathway enrichment analyses PPI network	Kaplan–Meier survival analysis Cox hazard model LASSO	VCAN-AS1, AL139002.1, LINC00326, AC018781.1, C15orf54	The five lncRNAs served as hub nodes in the ceRNA network.	[45]
64	GEO (GSE53137, GSE70880, GSE99417) TCGA-STAD	Sva and limma package miRDB, TargetScan, miRTarBase GO and KEGG pathway enrichment analyses PPI network	Kaplan–Meier analysis	UCA1, HOTTIP, HMGA1P4	Three lncRNAs may be involved in the development, and their prospective activities may be related to the prognosis of GC.	[41]
337	TCGA-STAD	GSEA-MSigDB database GO and KEGG pathway enrichment analyses	Kaplan–Meier survival analysis Log-rank test Cox and Lasso regression analysis	AL353804.1, AC010719.1, TNFRSF10A-AS1, AC005586.1, AL355574.1, AC009948.1, AL161785.1	These lncRNAs were enriched in the cell adhesion molecules and JAK-STAT3 signaling pathway.	[43]

Abbreviations: GC, gastric cancer; TCGA, The Cancer Genome Atlas; GEO, Gene Expression Omnibus; scRNA-seq, single-cell RNA sequencing; GSRCC, gastric signet ring cell carcinoma; TIME, tumor immune microenvironment; TIMER, Tumor Immune Estimation Resource; GSEA, gene set enrichment analysis; GO, gene ontology; KEGG, The Kyoto Encyclopedia of Genes and Genomes; PD, programming death; DEM, differentially expressed miRNA; MS, mass spectrometry; LASSO, least absolute shrinkage, and selection operator; GDSC, Genomics of Drug Sensitivity in Cancer; lncRNAs, long non-coding RNAs; ceRNA, competing endogenous RNA; GSVA, Gene set variation analysis; PPI, protein-protein interaction; STAT3, signal transducer and activator of transcription 3.

**Table 6 ijms-25-05880-t006:** Biomarkers for therapeutic prediction of GC using bioinformatical analysis.

Number of Patients	Databases	Analytical Platform	Statistical Analysis	Gene Set Identified	Comments	Refs.
253	GEO (GSE662254, GSE26942) TCGA CCLE	Limma R package GO pathway enrichment analyses	Cox regression analysis Kaplan–Meier method Log-rank test	*ND6*, *BRMS1*, *SRXN1*	Patients of the high-risk group should be treated with other chemotherapeutics to prevent unnecessary exposure to agents.	[69]
393	GC TCGA	MyCancerGenome CiViC TARGET OncoKB	*χ*^2^ test	*PIK3CA* mutation variants	Drugs identified here for therapeutic options are sorafenib, cabozantinib, regorafenib.	[70]
206	TCGA	iBAQ algorithm LC-MS/MS KEGG pathway enrichment analysis ssGSEA PRM method	Two-sided Student’s *t*-test) Kaplan–Meier survival curves Cox analysis	THSD4, SRPX2, TGFBI, THBS1, LAMB2, COL4A1, COL6A5, FN1, GP1BA, ITGA4, THBS3, THBS4, BCL2, BCL2L1, CASP3, CASP7 CDKN2A	PI3K-AKT signaling pathway had potential relevance to the ECM proteins.	[72]
804	GEO (GSE84437) TCGA GSEA TCIA	SVA package ConsensusClusterPlus package ssGSEA algorithm GO and KEGG enrichment pathways PPI network	Cox analysis Kaplan–Meier method	PLOD2, GLUT3	The lactate score may be useful for predicting the prognosis and immune escape of GC.	[71]
32	GEO (GSE1617297)	scRNA-seq CellChat R package UMAP algorithm BCR/TCR analysis	*t*-test, Wilcoxon rank-sum test,	CXCL13	CXCL13 is an important regulator for the immune response of the GC TIME.	[33]
47	Human Metabolome Database	LC-MS	*t*-test	Deoxyribose 1-phosphate, S-lactoylglutathione, lysoPC (16:0), and O-arachidonoyl ethanolamine	Four serum metabolites revealed potent divergences in patients with chemo-sensitive and chemo-resistant tumors.	[76]

Abbreviations: GC, gastric cancer; TCGA, The Cancer Genome Atlas; GEO, Gene Expression Omnibus; CCLE, cancer cell line encyclopedia; TCIA, The Cancer Immunome Database; iBAQ, intensity-based absolute quantification; ssGSEA, single sample gene set enrichment analysis; PRM, parallel reaction monitoring; PI3K, phosphoinositide 3-kinase; GSEA, gene set enrichment analysis; GO, gene oncology; KEGG, The Kyoto Encyclopedia of Genes and Genomes; TIME, tumor immune microenvironment; LC-MS, liquid chromatography-mass spectrometry.

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
