# Peer review of "Bioinformatics Analysis and Validation of Potential Markers Associated with Prediction and Prognosis of Gastric Cancer"

_ijms, 2024, doi:10.3390/ijms25115880_

Round 1

Reviewer 1 Report

Comments and Suggestions for Authors

This review presents a comprehensive summary of bioinformatics-based biomarkers for the prognosis of gastric cancer. It provides readers with the latest biomarkers for GC and new methods for analyzing high throughput data. The authors also emphasize biomarker discovery based on multi-omics datasets, such as genetic, epigenetic, transcriptomic, proteomic, and metabolomic signatures. They discussed the future impact of multi-omics in clinical utility, which is an interesting study. It would be better if the following minor concerns could be addressed.

1.    There are descriptions of “… predictive and prognostic biomarkers …” in the manuscript.  What’s the difference between predictive biomarkers and prognostic biomarkers?  Predictive markers can be used to predict the prognosis, and therapeutic effects, or even identify individuals at high risk of GC (for diagnosis).

2.    It’s unclear whether patients in each study were paired. Also, the characteristics such as age, race, and stage of patients from different databases are not clear. Therefore, normalization of GC data from different databases is very important to integrate multi-omics data. When it comes to the clinical utility of the presented biomarkers, they still need to be validated through comprehensive cohort studies and large-scale clinical trials.

3.    Table 1 contains both genomic-based and epigenomic-based biomarkers, so the Title of Table 1 may not be precise. The column on Survival analysis contains not only Survival Analysis but also t-test, chi-square test, et. al., statistical analysis? The way to present t-tests is not consistent in different Tables. Also, the definition of the “validation methods” column in the five Tables is not clear. For example, the validation method for proteins is PCR in Table 3 (citation 46)?

Author Response

Response to reviewers

Reviewer #1:

  1. There are descriptions of “… predictive and prognostic biomarkers …” in the manuscript.  What’s the difference between predictive biomarkers and prognostic biomarkers?  Predictive markers can be used to predict the prognosis, and therapeutic effects, or even identify individuals at high risk

Response: We appreciate the reviewer’s comments. In this review, we focus on the role of predictive biomarkers in predicting therapeutic strategies. According to the reviewer’s suggestion, we added a paragraph on the meaning of predictive biomarkers in the text on rows 536-539.

  1. It’s unclear whether patients in each study were paired. Also, the characteristics such as age, race, and stage of patients from different databases are not clear. Therefore, normalization of GC data from different databases is very important to integrate multi-omics data. When it comes to the clinical utility of the presented biomarkers, they still need to be validated through comprehensive cohort studies and large-scale clinical trials.

Response: We appreciate the reviewer’s comments. In compliance with the reviewer’s suggestion, we added sentences on rows 637-640.

  1. Table 1 contains both genomic-based and epigenomic-based biomarkers, so the Title of Table 1 may not be precise. The column on Survival analysis contains not only Survival Analysis but also t-test, chi-square test, et. al., statistical analysis? The way to present t-tests is not consistent in different Tables. Also, the definition of the “validation methods” column in the five Tables is not clear. For example, the validation method for proteins is PCR in Table 3 (citation 46)?

Response: We appreciate the reviewer’s comments. We have checked the Table carefully and revised as possible. We divided Table 1 into Table 1 and Table 2 and made a new title. We changed “Survival Analysis” to “Statistical Analysis” and omitted the “validation methods” column,

Reviewer 2 Report

Comments and Suggestions for Authors

The overall goal of this review manuscript is to summarize the recent trends on how analysis using bioinformatics contributes to biomarker discovery for prediction and prognosis of therapeutic efficacy in gastric cancer (GC). This manuscript is well written and very comprehensive and will be valuable to readers especially in the area of GC. One comment I probably have is to try to fit the table in a single page if possible. I think if this is not possible, just include as part of supplementary material. In the abstract, I think it’s important to summarize how many studies were evaluated and what are the key findings. This review is very broad but I am wondering if there is a way to highlight the most salient points so readers can have a focus.

On page 9, the first two lines have spaces; these spaces should be removed. The study is very comprehensive and detailed and I think will attract many readers.

Comments on the Quality of English Language

I think the English language is acceptable overall.

Author Response

Response to reviewers

Reviewer #2:

  1. One comment I probably have is to try to fit the table in a single page if possible. I think if this is not possible, just include as part of supplementary material.

Response: We appreciate the reviewer's comments. According to the reviewer’s suggestion, we try to resize the tables to fit on a single page. We divided Table 1 into Tables 1 and 2 and omitted the optional studies.

  1. In the abstract, I think it’s important to summarize how many studies were evaluated and what are the key findings. This review is very broad but I am wondering if there is a way to highlight the most salient points so readers can have a focus.

Response: We appreciate the reviewer's comment. In compliance with the reviewer’s suggestion, we added, "The practical integration of bioinformatics analysis and multi-omics datasets under complementary computational analysis is having a great impact on the search for predictive and prognostic biomarkers and may lead to an important revolution in treatment.” in the abstract to highlight the most salient points.

  1. On page 9, the first two lines have spaces; these spaces should be removed.  

Response: We appreciate the reviewer's comments. We removed spaces according to the reviewer’s recommendation.